# Dynamic bistable switches enhance robustness and accuracy of cell cycle transitions

**Jan Rombouts** ᴼ *, **Lendert Gelens** ᴼ *

Laboratory of Dynamics in Biological Systems, Department of Cellular and Molecular Medicine, University of Leuven (KU Leuven), B-3000 Leuven, Belgium

* jan.rombouts@kuleuven.be (J.R.); lendert.gelens@kuleuven.be (L.G.)

## Abstract

Bistability is a common mechanism to ensure robust and irreversible cell cycle transitions. Whenever biological parameters or external conditions change such that a threshold is crossed, the system abruptly switches between different cell cycle states. Experimental studies have uncovered mechanisms that can make the shape of the bistable response curve change dynamically in time. Here, we show how such a dynamically changing bistable switch can provide a cell with better control over the timing of cell cycle transitions. Moreover, cell cycle oscillations built on bistable switches are more robust when the bistability is modulated in time. Our results are not specific to cell cycle models and may apply to other bistable systems in which the bistable response curve is time-dependent.

**Data Availability Statement:** The code used to generate the computational results can be found at https://github.com/JanRombouts/dynamicswitches.

## Author summary

Many systems in nature show bistability, which means they can evolve to one of two stable steady states under exactly the same conditions. Which state they evolve to depends on where the system comes from. Such bistability underlies the switching behavior that is essential for cells to progress in the cell division cycle. A quick switch happens when the cell jumps from one steady state to another steady state. Typical of this switching behavior is its robustness and irreversibility. In this paper, we expand this viewpoint of the dynamics of the cell cycle by considering bistable switches which themselves are changing in time. This gives the cell an extra layer of control over transitions both in time and in space, and can make those transitions more robust. Such dynamically changing bistability can appear very naturally. We show this in a model of mitotic entry, in which we include a nuclear and cytoplasmic compartment. The activity of a crucial cell cycle protein follows a bistable switch in each compartment, but the shape of its response is changing in time as proteins are imported into and exported from the nucleus.

**Funding:** This work was supported by the Research Foundation Flanders (FWO, www.fwo.be) with individual support to J.R. and project support to L.G. (Grant GOA5317N) and the KU Leuven Research Fund (No. C14/18/084) to L.G. The computational resources and services used in this work were provided by the VSC (Flemish Supercomputer Center), funded by the Research Foundation Flanders (FWO) and the Flemish Government - department EWI. The funders had no role in study design, data collection and analysis, decision to publish, or preparation of the manuscript.

**Competing interests:** The authors have declared that no competing interests exist.

# 1 Introduction

Multistability is one of the clearest manifestations of nature's nonlinearity. A multistable system can, under exactly the same conditions, be in different stable steady states. Consider a ball moving on a hilly terrain under the influence of gravity, where every valley corresponds to a stable state for the ball (Fig 1A). When there are multiple valleys, the ball's initial position determines where it will end up. These valleys can appear and disappear as the shape of the terrain changes. Another way to look at such a changing terrain is by plotting the steady state position of the ball (labeled output) as a function of a parameter that determines the shape of the terrain (labeled input) (Fig 1B). Here, for low input, there is only one steady state (situation 1). By increasing the parameter, a new state appears and the system is said to be bistable (situation 2). When the input crosses a threshold value, the initial stable valley disappears and the ball is forced to move to the right valley (situation 3). This transition is discontinuous, fast and irreversible. In between two stable states, there is an unstable steady state (the maximum in Fig 1A and the dashed line in Fig 1B). The points at which the stable and unstable steady state coalesce define the threshold values. These points are also called saddle-node points in the language of bifurcation theory.

This simple mechanical example has equivalents in all sorts of physical and biological systems, where Newton's laws of motion and the hilly terrain are replaced by chemical reactions, predator-prey interactions, heat transport or other mechanisms. In climate and ecology studies, transitions to a new steady state are often called tipping points [1, 2], and they are of special interest given current climate change. Bistability is present on all scales, ranging from the global climate system [2] to a single cell [3]. The genetic system involving the *lac* operon in an *E. coli* cell [4] allows bacteria to switch between using glucose or lactose. Bistability in its actin dynamics enables a cell to quickly and robustly switch between different migration modes [5]. Besides these *in vivo* examples, bistable responses have also been observed in purified kinase-phosphatase systems [6, 7] and are a common objective in the design of synthetic genetic systems [8]. The concept of bistability and irreversible transitions also plays an important role in cell differentiation. There, the image of balls rolling down valleys is echoed in Waddington's epigenetic landscape [9].

On the molecular level, bistability is generated by the interplay of a large amount of molecules involved in chemical reactions. The conditions under which these reaction networks generate bistability have been extensively studied. Typically, one needs highly nonlinear

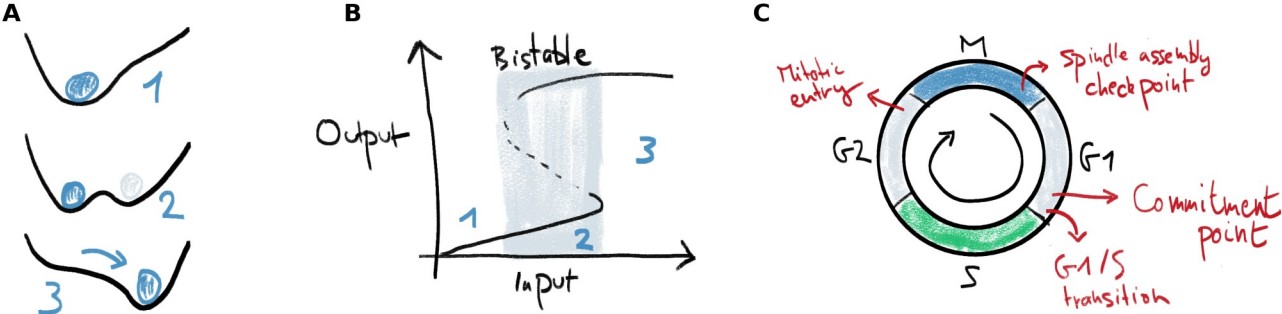

**Fig 1. Bistability allows robust switching and is common in the cell cycle.** A) A clear example of bistability in a dynamical system occurs when a little ball moves under the influence of gravity on a hilly terrain. Valleys correspond to stable steady states. These can be created and destroyed under influence of an external parameter. When a steady state disappears, the ball quickly transitions to another steady state. B) Representation of the ball's position as function of a parameter which determines the shape of the terrain in Panel A. When the input increases beyond a threshold, the left equilibrium position in Panel A disappears and the ball quickly moves to the other stable position. C) In the cell cycle, bistable switches underlie some of the important transitions and checkpoints.

(ultrasensitive) responses and positive feedback loops [10, 11]. However, bistability can also be present in simple systems with a minimal amount of components governed by mass-action kinetics. Finding the conditions under which such systems generate multistability is one of the important questions asked in chemical reaction network theory, where mostly algebraic methods are used to analyze these systems (eg. [12–15]).

To survive and proliferate, a cell has to replicate its DNA and structural components, and then distribute this material evenly to its daughters. This process is governed by the orderly progression through different phases of the cell cycle. The eukaryotic cell cycle contains various checkpoints and transitions in which bistability plays a role (Fig 1C), and can even be viewed as a chain of sequentially activated bistable switches [16–19]. These switches provide robustness and directionality to the cell cycle and ensure the genome's integrity. Both the "commitment point", where a cell becomes committed to enter the cell cycle, and the transition from G1 to S phase have been associated to underlying bistable switches [20–23]. Later, after the cell has duplicated its DNA, there is a sudden transition from G2 to mitosis, characterized by the prompt activation of cyclin-dependent kinase 1 (Cdk1). This sudden mitotic entry has also been shown to be controlled by two bistable switches [24–28]. Further in mitosis, the spindle assembly checkpoint (SAC) controls the correct separation of sister chromatids at the metaphase-anaphase transition [29]. Theoretical models have shown that there are molecular mechanisms that can lead to bistability underlying this checkpoint [30–32].

The standard view of these cell cycle transitions is that the two states are represented by two branches of a static bistable response curve. The transition happens when a slowly changing input reaches a threshold, upon which the system jumps to the other branch of the curve (Fig 1B). Throughout this work we will focus on the bistable switch in mitotic entry, arguably the cell cycle switch that has been best characterized experimentally and theoretically. Already in the early 1990s, mathematical models showed how biochemical interactions could lead to cell cycle oscillations that switched between interphase and mitosis [33, 34], and later predicted that bistability might be at the basis of the mitotic entry transition [35, 36]. This bistability was later verified experimentally [24, 25]. Briefly, the biochemical interactions generating the switch are as follows. The kinase Cdk1 becomes active when bound to a Cyclin B subunit, and is involved in two feedback loops: Cdk1 activates the phosphatase Cdc25, which removes an inhibitory phosphorylation on Cdk1, thereby activating it and closing a double positive feedback loop. Secondly, Cdk1 inhibits Wee1, a kinase responsible for inhibiting Cdk1 through phosphorylation. This constitutes a double negative feedback loop. Due to ultrasensitivity in these feedbacks, a bistable response of Cdk1 activity to Cyclin B concentrations is generated. The shape of this switch depends, among other factors, on the amounts of Wee1 and Cdc25 present [37] (for more details, see Section 2.1).

In many cell cycle transitions, the response curve is not static, since the parameters which determine the shape of the switch are changing, either slowly or in a more sudden fashion. Typically the shape of the bistable response curve depends on the total concentration of proteins implicated in the feedback loops. These concentrations may change, either due to production and degradation, or due to relocalization of proteins in space. We can consider the cell as a set of compartments with slow fluxes between them. If each compartment is well mixed, it has its own bistable response curve. The shape of this curve depends on the concentrations of proteins in that compartment, which can change over time as proteins relocalize. Note that compartmentalization has been studied in the context of bistability already: adding different compartments can be a mechanism of generating a bistable response, where there is none in a single well-mixed system [38, 39]. In Section 2.1, we discuss how considering nucleus and cytoplasm as interacting compartments can alter the bistable switches governing mitotic entry.

The importance of dynamically changing the bistable response curve has been acknowledged before, mostly in the context of lowering an activation threshold. For example, the threshold for mitogenic signaling, which defines the commitment point, can be influenced by DNA damage, cell volume or cell contacts [23, 40]. This provides extra control over the timing of passing the commitment point. At mitotic entry, Wee1 is known to be quickly degraded [41, 42], which lowers the threshold for Cdk1 activation [43] and triggers a transition into mitosis. Moreover, Wee1 is regulated by the 24h circadian clock (see e.g. the review in ref. [44]), which suggests that changing activation thresholds might be a mechanism by which the circadian rhythm influences the cell cycle.

Bistability also lies at the heart of an important class of oscillations which appear time and again in chemical, biological and physical systems. These oscillations are called relaxation oscillations, and consist of slow progress along the branches of a bistable system, with sudden jumps between them. Eminent examples of relaxation oscillators are the Van der Pol and Fitz-Hugh-Nagumo type systems. Whereas they were developed as models of electrical systems, either engineered, or in neurons, now they are often used as generic oscillating systems which can exhibit different kinds of dynamics [45].

Nonlinear oscillators generate many periodic phenomena in cell biology, among which circadian rhythms, metabolic oscillations, and also the early embryonic cell cycle (see the books [46, 47] and review papers [48, 49] for overviews of biological oscillations). Although bistability is not essential to generate oscillations, it has been shown that the addition of positive feedback, which can lead to bistability, can endow biological oscillations with a more stable amplitude [50]. The early embryonic cell cycle—in contrast to the somatic cycle—largely lacks checkpoint control, gap phases and even growth. The cycle is driven forward as a true oscillator by periodic production and degradation of proteins.

Here, we investigate how dynamically changing bistable switches affect transitions and relaxation oscillations. To motivate our studies, we first show how including different cellular compartments in a model of mitotic entry leads naturally to a situation with bistable switches that change in time. Next, we explore the concept of dynamically changing switches using a simple model. After introducing the model, we discuss how a single transition, such as the crossing of a cell cycle checkpoint, is affected by dynamically changing the activation threshold and the shape of the response curve. We demonstrate that in a noisy system, a dynamically changing switch confers robustness to the transition timing if the noise in the slow variable (the input, as we called it) is negligible compared to the noise in the fast variable (the output). We also describe a mechanism which may be at play in spatially-extended systems. There, bistability can lead to traveling fronts, whose speed depends on the shape of the bistable response curve. Front propagation can therefore dynamically change as proteins—which determine the shape of the response curve—are redistributed in space. We then discuss how such a dynamically changing switch affects relaxation oscillations. We show how the period of these oscillations is made more robust to noise under similar conditions as for a single transition. A changing switch also enlarges the parameter region in which oscillations occur. We interpret our general results in the context of mitotic entry, and in the discussion we consider how dynamically changing bistability might be used in interpreting other models.

## 2 Results

### 2.1 A model for mitotic entry shows how dynamic switches appear in two cellular compartments

Mitotic entry is triggered by the activation of the kinase Cdk1, which sets into motion many of the changes a cell undergoes during mitosis. Cdk1 becomes active when bound to a Cyclin B

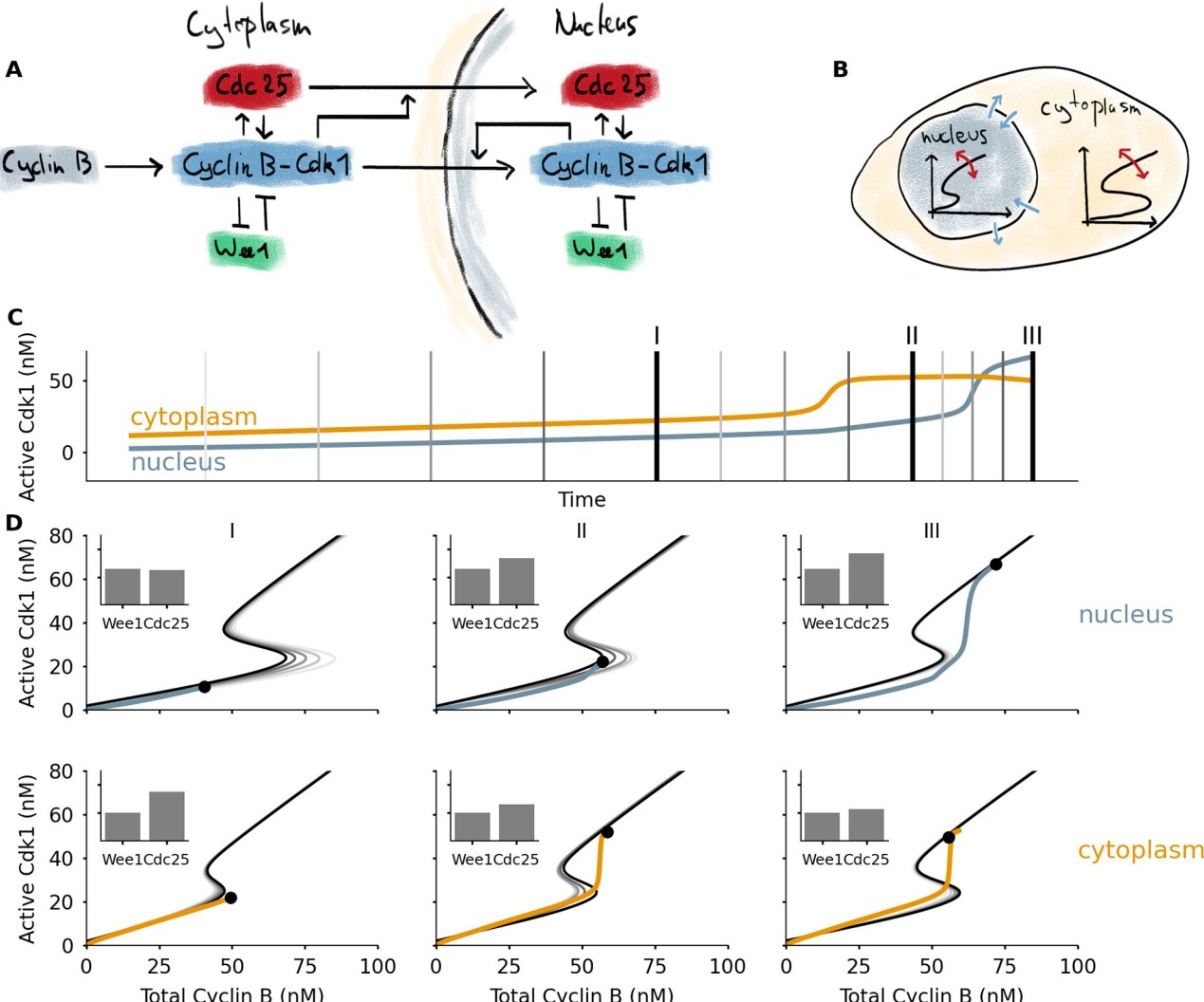

**Fig 2. A model for mitotic entry shows how dynamic switches appear in two cellular compartments.** A) The core protein interaction network involved in mitotic entry with double positive and double negative feedback loops centered on the Cyclin B-Cdk1 complex. Additional feedbacks act on the import rates of Cdc25 and Cyclin B-Cdk1. B) Two cellular compartments can exchange proteins through import and export. The total concentration of Cdc25 in each compartment determines the shape of the bistable response, which changes over time. C) Time series of active Cdk1 in nucleus and cytoplasm, from a simulation of mitotic entry driven by Cyclin B production in the cytoplasm. The vertical lines correspond to the different response curves in Panel D. D) Left: initially the activation threshold in the nucleus lies to the far right, due to the dominance of Wee1 over Cdc25 there. The activation threshold shifts to the left as Cdc25 is imported, which happens faster as Cdk1 activity rises in the cytoplasm. Middle: the activation threshold for Cdk1 activation is first crossed in the cytoplasm. The sudden jump in Cdk1 activation effects a sudden increase of Cdc25 import into the nucleus, which in turn quickly lowers the activation threshold there. Right: the decrease of the threshold in the nucleus triggers activation of Cdk1, leading to additional import of Cyclin B and a high Cdk1 activity. The black dot denotes the position of the system, the black curve corresponds to the bistable response at the time point corresponding to the dot. The gray lines are snapshots of the bistable response at times leading up to this point, corresponding to the time points indicated in Panel C. An animation which more clearly illustrates the dynamics can be found in S4 Video.

subunit. Additionally, Cdk1 activity is controlled by its phosphorylation state, which is regulated by the kinase Wee1 and the phosphatase Cdc25 (Fig 2A). In turn, Cdk1 itself activates Cdc25 and inactivates Wee1. These feedback loops produce a bistable response of Cdk1 activity as function of total Cyclin B levels [24, 25]. The different feedback loops have been characterized in detail [51, 52]. Mitotic entry involves many other mechanisms, such as a phosphatase switch [26, 27] or the regulation of other kinases such as those from the Polo or

Aurora families. An excellent recent review of the mitotic entry transition is given by Crncec and Hochegger [53].

One particular source of additional regulation is the spatial localization of the different proteins. In mitosis the Cyclin B-Cdk1 complexes accumulate in the nucleus [38, 54]. Cdc25 also translocates to the nucleus at mitotic entry [55]. Wee1, the kinase inhibiting Cdk1, is mostly nuclear during interphase [56], possibly to make sure that Cdk1 is not activated too early, i.e. before DNA replication—which takes place in the nucleus—is complete. Spatial regulation of other mitotic regulators such as Polo [57] and Greatwall [58, 59] has recently been shown to be important for correct progress of mitosis as well.

All of these spatial translocations influence the behavior of the system, and here we show that this can be interpreted in the framework of a bistable switch with dynamically changing shape. To this end we extend the cell cycle model of Yang and Ferrell [60] to include two different compartments: the nucleus and the cytoplasm. In each compartment, Cdk1 activation is governed by the feedback loops through Wee1 and Cdc25. In addition, proteins can move into and out of the nucleus. To include spatial feedback, the nuclear import rates may depend on the concentrations of active Cdk1 [38]. In our simplified version, we assume that active cytoplasmic Cdk1 enhances import of Cdc25 and active nuclear Cdk1 enhances import of Cyclin B. These assumptions are approximations of the experimentally known feedbacks [38, 55, 57]. We assume that Wee1 concentrations are higher in the nucleus. If the import and export rates are slow relative to the activation dynamics of Cdk1, we can consider each compartment to be nearly in steady state. This steady state, in turn, follows the bistable response curve of Cdk1 as a function of total Cyclin B. Due to translocation of Cdc25, the shape of these curves varies (Fig 2B). We do not aim to include all of the complexity of mitotic entry described in the previous paragraph. Rather, we want to show how including a minimal spatial component using plausible mechanisms leads to changed mitotic entry dynamics, which can be interpreted using dynamic bistable switches. More details and the full set of equations used can be found in the Methods section.

Adding these compartments leads to a mitotic entry in different steps (Fig 2C and 2D). First, Cyclin B accumulates in the cytoplasm. At the start, the threshold for Cdk1 activation is lower in the cytoplasm than in the nucleus, due to the lower Wee1 concentration there. As a consequence, Cdk1 activation occurs first in the cytoplasm. This activation triggers the import of Cdc25 into the nucleus, which lowers the activation threshold there and allows Cdk1 activation in the nucleus. This triggers a translocation of even more Cyclin B to the nucleus. These two effects ensure that the activation of Cdk1 in the nucleus is very fast, irreversible, and happens after cytoplasmic activation of Cdk1. Once Cdk1 is activated in the nucleus, nuclear import of Cyclin B stays high. Most newly synthesized Cyclin B will be imported in the nucleus, further raising Cdk1 activity levels. Cdk1 activity in the cytoplasm settles at a nearly constant value. The animation S4 Video makes the time evolution of the different switches more clear.

The key observation we want to stress with this illustration is that the activation threshold is different in nucleus and cytoplasm, and importantly, that this threshold is controlled by translocation of Cdc25. By spatially regulating Cdc25, the cell has an additional layer of control over the timing of Cdk1 activation. The combined feedbacks lead to quick Cdk1 activation in the nucleus, and an enhanced import rate makes sure that Cdk1 activity in the nucleus increases further.

The translocation of Cdc25 is only one example of a mechanism which might produce changing bistability. Another candidate where this viewpoint can apply is nuclear envelope breakdown (NEBD). Cdk1 activation triggers this event [61] which effectively mixes the two compartments. In turn, the two bistable response curves collapse to a single one. Yet another situation where changing bistability might be at play is the translocation of other proteins, such as the kinase Greatwall. This will likely have an effect on the shape of the second bistable switch in mitosis [26, 27]. The inclusion of these different mechanisms in a larger model of mitotic entry,

and their interpretation using dynamically varying switches, is an avenue for further study. We would like to remark that the effect of Greatwall and NEBD on the bistable response curves has been studied already in the context of mitotic collapse [62].

## 2.2 A simple model shows that transition timing is more robust and accurate in a system with a dynamic switch

The results of the previous section show that a dynamically changing switch can appear naturally in a biochemical system. To investigate the consequences of such a changing switch in more depth, we introduce a simple model. The model describes a protein which can be in an active or inactive state (Fig 3A). The protein is involved in two feedback loops: it promotes its own activation and inhibits its inactivation. We model this using the equation

$$\frac{dX}{dt} = f(X)(X_T - X) - g(X)X, \tag{1}$$

where $X$ is the concentration of active protein and $X_T$ is the total amount of this protein, $X_T = X + X_{\text{inactive}}$. The functions $f$ and $g$ are given by

$$\begin{aligned}
f(X) &= a + b\frac{X^n}{K^n + X^n}, \\
g(X) &= a' + b'\frac{K'^m}{K'^m + X^m}.
\end{aligned} \tag{2}$$

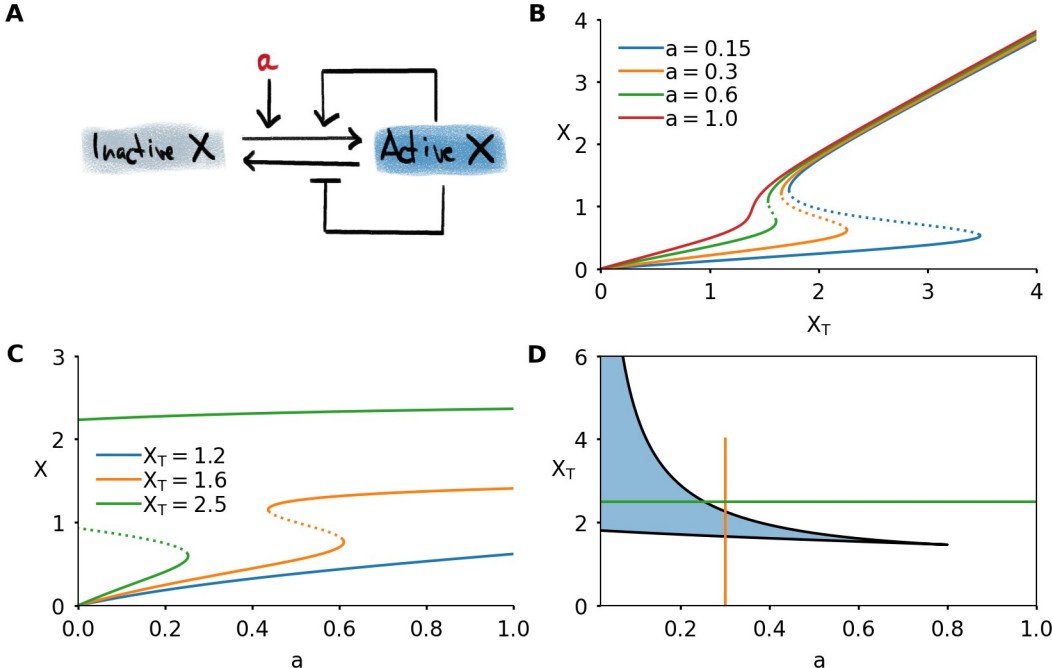

**Fig 3. A simple model of protein activity shows bistability.** A) Interaction diagram of a model consisting of single protein which can be active or inactive. Active protein promotes its own activation and inhibits its inactivation. The basal activation rate is given by the parameter $a$. B) Steady state response to total protein $X_T$, for different values of $a$. C) Steady state response to basal activation rate $a$, for different values of total concentration $X_T$. D) Two-parameter bifurcation diagram of the system. The bistable region is shaded. The vertical orange curve, when followed from bottom to top, corresponds to the orange response curve in Panel B. The horizontal green curve, followed from left to right, corresponds to the green response curve in Panel C. Parameter values not mentioned in the plots can be found in Table 1 in the Methods section.

These Hill functions, which we use to model positive and negative feedback, are typically the outcome of basic biochemical reactions that generate ultrasensitivity, such as substrate competition, multisite phosphorylation, or others [63].

The combination of the different feedback loops and the steep response functions is known to generate bistability [11]. Indeed, this system shows bistable behavior, which can be visualized in different ways (Fig 3B and 3C). The steady state response of the active protein level $X$ can be bistable as function of the total amount of protein $X_T$ (Fig 3B). The shape of this response curve depends on the value of $a$, the basal activation rate of the protein. High values of $a$ correspond to high basal activation of $X$. This ensures that any protein in the system will be directly converted into its active form, and $X$ increases nearly linearly with $X_T$. For low values of $a$, there is bistability, and the activation threshold becomes higher with decreasing $a$. As a consequence, for very low $a$ a large amount of protein needs to be added to the system to initiate the feedback loops that will lead to a full activation of the protein. If $X_T$ is continuously increased, for example through a constant production of protein in the inactive state, at a certain moment the threshold will be reached and the system will jump to the active state.

This representation is closely related to the cell cycle control system for mitotic entry, where the total abundance of Cyclin B ($\sim X_T$) gradually increases until the threshold for mitotic entry is reached and Cdk1 ($\sim X$) is activated [24, 25, 35]. In this scenario, the basal activity of the phosphatase Cdc25 plays the role of the parameter $a$ [37, 52].

A different bistable response emerges when plotting $X$ as a function of $a$, keeping $X_T$ fixed (Fig 3C). The shape of the response curve now depends on the value of $X_T$. For low values of $X_T$, there is no bistability and the response is approximately hyperbolic. Intermediate levels of $X_T$ lead to a bistable response curve. For high levels of $X_T$, the switch is not only irreversible for small changes in the input, but even lowering the input to zero is not sufficient to drive the system back to its inactive state. This happens because the left threshold occurs at $a < 0$. Since in biological systems $a$ corresponds to an activation rate, a positive quantity, this makes it impossible for the system to go back to its inactive state. In such an irreversible switch, the system can transition from the low to high state, but it cannot go back. Dynamically varying $X_T$ would provide a solution: by controlling the levels of $X_T$, the transition back to the low state can be made possible. We have previously explored this mechanism in a model of the interaction between Cdk1 and the protein kinase Aurora B, which plays an important role during chromosome segregation in mitosis [64].

The effect of $a$ and $X_T$ can be summarized in a two-parameter bifurcation diagram (Fig 3D). A response curve where only one of the two parameters is varied (Fig 3B and 3C) corresponds to a horizontal or vertical cut in this diagram. In the remainder of this work, we will focus on the situation as in Fig 3B: $X_T$ is the main parameter—the input, as we previously called it—and we will explore the effects of having either a constant value of $a$ or a dynamically changing $a$.

In order to study the transition from low to high activity when protein is produced, we study the following system of equations:

$$
\begin{aligned}
\frac{dX}{dt} &= \varepsilon^{-1}(f(X)(X_T - X) - g(X)X) \\[6pt]
\frac{dX_T}{dt} &= k_X,
\end{aligned}
\tag{3}
$$

where the second equation corresponds to a constant increase in protein abundance $X_T$. The small parameter $\varepsilon$ is added to model timescale separation: the activation-inactivation dynamics of the protein are much faster than its production. As the total concentration increases, the

system moves along the bottom branch of the bistable response curve. When the concentration crosses the activation threshold, the protein is rapidly activated (Fig 4A).

We now set out to investigate how this activation is affected when the shape of the bistable switch is changing while $X_T$ is increasing. We impose the following functional form on $a$ (Fig 4B):

$$a = \bar{a} + \Delta a \tanh(\kappa(X_T - X_c)). \tag{4}$$

The value of $\bar{a}$ is the value around which $a$ varies symmetrically. By tuning the parameter $\Delta a$ we can control the extent of the shape changes: for $\Delta a = 0$, the switch does not change and $a = \bar{a}$ is a constant. For $\Delta a = \bar{a}$, $a$ varies between extremes of 0 and $2\bar{a}$. The parameter $\kappa$ controls the abruptness with which the bistable shape changes when $X_T$ crosses a threshold $X_c$. This threshold is chosen to be the midpoint of the folds of bistable switch. For positive values of $\Delta a$, the activation threshold of the switch moves to the left while $X_T$ increases. The dynamics of such a system are illustrated in Fig 4C and in S1 Video.

One striking consequence of the dynamic switch is that the level of $X$ is kept very low until its activation, whereas if $a$ is not changing, $X$ already increases while the system is approaching the threshold. Moreover, by lowering the activation threshold while the system is approaching the transition point, the timing of activation can be controlled more precisely. To illustrate this, we implement stochastic versions of the model. First, we study a Langevin equation by adding noise with magnitude $\sigma$ to the $X$ variable (see Methods for details). In this system, noise can trigger the system to jump to the high activity state even before the activation threshold is reached. We simulate the system many times and measure the transition time, defined as the time the value of $X$ crosses a threshold (Fig 4D). A dynamic switch shows less variation in the transition time than a static switch (Fig 4E). Increasing the amplitude (higher $\Delta a$) and abruptness (higher $\kappa$) of the dynamical shape changes help to further decrease the variation in transition timing (Fig 4F and 4G).

The Langevin equation is an approximation of real biological sources of noise. We also tested a fully stochastic system, using the Gillespie algorithm [65] with three reactions: activation, inactivation and production. The noise magnitude $\sigma$ is replaced here by a system size $\Omega$, which correlates with the number of molecules in the system. Details can be found in the Methods section.

The main difference with the Langevin version is the fact that noise now also appears in the production, such that $X_T$ does not increase smoothly (Fig 4H). In this system, the effect of the changing switch is still observable, but much smaller (Fig 4I and 4J). We had to take more samples of the system to observe the decrease in variation. This decreased effect might be due to the implementation of the changing switch: the value of $a$ depends directly on the value of $X_T$ (Eq 4). If $X_T$ is noisy, the switch variation itself is noisy, which mitigates the stabilizing effect the switch variation had on the transition time in the Langevin version of the model. Moreover, our measure of the accuracy is the coefficient of variation, defined as standard deviation divided by the mean. Whereas the standard deviation clearly decreases for larger switch variations, the mean also goes down, leading to a less clear effect on the coefficient of variation. S1 Fig shows plots of standard deviation, mean and CV. Overall, we conclude that a dynamic switch makes transitions more robust in our simple model if the noise in the slow variable is much smaller than the noise in the fast variable. An interesting future direction is to study the effect of noise in a more realistic system, in which the changing switch is not implemented artificially like here, but arises from a biological mechanism. Perhaps the requirement on the magnitude of the noise can be relaxed in such a model.

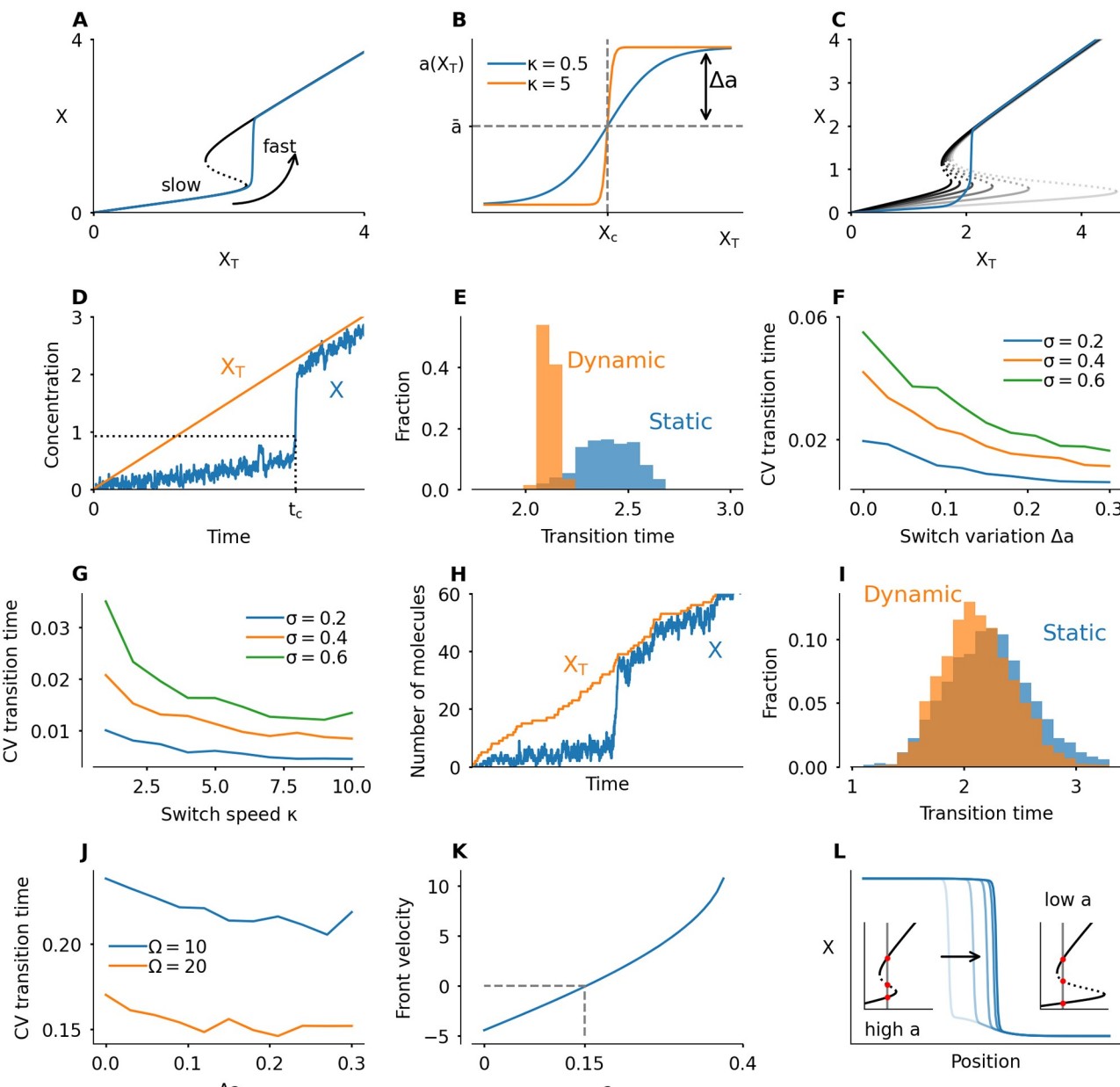

**Fig 4. A dynamically changing bistable switch enhances robustness and accuracy of transitions in time and space.** A) When $X_T$ increases at a constant rate (here $k_X = 0.2$), the activity of the protein will increase suddenly at the moment $X_T$ crosses the activation threshold of the switch. B) Function used to make the shape of the switch dependent on the total amount of protein, by coupling $a$ to $X_T$. The switch variation depends on the value of $\Delta a$, which controls the magnitude of possible deviations of $a$ from a mean value $\bar{a}$. The parameter $\kappa$ controls how abruptly the system switches between low and high $a$ values. C) Time evolution of a system in which $X_T$ increases at a constant rate, and $a$ is coupled to $X_T$. The gray response curves are snapshots in time. The activation threshold starts out to the far right, and moves left as $X_T$, and with it $a$, increases. Here $\bar{a} = 0.3, \Delta a = 0.2, \kappa = 5, k_X = 0.2$. D) Evolution of a system in which noise is added to the $X$ variable (Langevin version). The transition time $t_c$ is defined as the time when $X$ crosses a threshold value. Here $\bar{a} = 0.3, \Delta a = 0, \kappa = 5, k_X = 0.2, \sigma = 0.6$. E) Histogram of measured transition times for a static switch ($\Delta a = 0$) and a dynamic switch ($\Delta a = 0.3$), with $\bar{a} = 0.3, \kappa = 5, k_X = 1, \sigma = 0.6$. The spread is lower for the dynamic switch. F) Coefficient of variation (CV), defined as standard deviation divided by mean, of the transition time, as function of the switch variation $\Delta a$. Here $\bar{a} = 0.3, \kappa = 5, k_X = 1$. G) Coefficient of variation as function of $\kappa$, which defines the speed by which $a$ changes. Faster changing corresponds to smaller deviations. Here $\bar{a} = 0.3, \Delta a = 0.3, k_X = 1$. H) Time series of a run of the Gillespie algorithm/stochastic simulation algorithm (SSA). Parameter values are the same as in panel D, but here molecule counts are discrete. We used a system size $\Omega = 20$. I) Histogram of transition times in the stochastic simulation algorithm (compare to panel E) with $\Omega = 20$. The variation is lower for the dynamic switch, but the difference is much smaller than in the Langevin version. J) Coefficient of variation as function of $\Delta a$ for the stochastic simulation algorithm (compare to Panel F). K) Velocity of a bistable front as function of $a$, for $X_T = 2$. A positive velocity means that the active protein state overtakes the inactive state (the front shown in panel L moves to the right). At $a \approx 0.15$, the front is stationary. L) A bistable front in the presence of an inhomogeneous $a$ profile in space. On the left, $a = 0.27$,

which means the front moves to the right. On the far right, $a = 0.13$ is low, which means that the front moves to the left. The result is that the front is pinned in the middle where $a \approx 0.15$. This pinning can be lifted by a redistribution of $a$ (see S2 Video). Other parameters for all panels can be found in the Methods section, Table 1. Histograms and coefficients of variations where calculated over 200 simulations for the Langevin version (panels E to G) and 2000 simulations for the SSA (H to J).

Noise is inevitable in biochemical systems, and can be a nuisance or something the cell uses to its advantage [66]. Here, in the context of the cell cycle, premature activation due to noise is to be avoided. We conclude that accurate control of the timing of transition can be achieved by dynamically changing the switch and increasing $a$ as $X_T$ approaches the threshold.

## 2.3 Transitions in space can be controlled by dynamically changing the bistable switch

When bistable systems are coupled in space in the presence of diffusion, they may produce traveling fronts. In our model with active and inactive protein, a traveling front can arise when one region of space has a high $X$ activity and an adjacent region has low activity. The interface between these regions starts to move, depending on which state is dominant. Such traveling fronts are omnipresent in biology, where they usually have a signaling or synchronizing function [45, 67].

The speed and direction of traveling fronts depend on the parameters of the system. Consider for example a traveling front that links regions of high and low activity of the protein $X$, with $X_T = 2$ fixed. The direction of the front depends on $a$: low $a$ corresponds to a dominant low activity state, high $a$ to a dominant high activity state. The front moves such that the dominant state overtakes the other one (Fig 4K).

Let us assume that the parameters may vary in space, such that the front speed itself varies in space. Consider a system where $a$ is high in one region, low in another and has a smooth transition between both regions. In this case, the front would move to the right until it hits the transition region, where it slows down and comes to a halt (Fig 4L). This phenomenon is called pinning. Front pinning and localization—possibly due to a spatial inhomogeneity, as here— occur frequently in physical systems [68]. In biology, front pinning mechanisms have been studied in the context of cell polarization [69], and in ecosystem transitions [70].

The front comes to a halt due to a spatially heterogeneous profile of $a$. A pinned front can then be released by redistributing $a$, and thus changing the bistable switch and the dominant state (see S2 Video). Dynamically changing the parameters which affect the shape of the bistable response curve can thus provide the cell with extra control over spatial transitions.

Waves of Cdk1 activity spread throughout the cell at mitotic entry, as has been observed in *Xenopus* cell-free extracts [71–73] and in the early *Drosophila* embryo [74, 75]. In the cell, and in extracts, spatial heterogeneities are present through nuclei, which concentrate certain proteins [72]. In such systems, the effect of dynamic bistability on front dynamics is likely present, all the more because the spatial heterogeneity drastically changes at nuclear envelope breakdown. In *Drosophila*, dynamic changes in the bistable switch have been shown to play an important role in determining the nature of mitotic waves [75].

## 2.4 A dynamic switch promotes stable oscillations

Fast transitions between states form the basis of relaxation oscillations, such as those observed in early embryonic cell cycles of *Xenopus laevis*, where the cell quickly switches between interphase and mitosis. We investigate how such oscillations are affected by a dynamic bistable switch. We expand our model to obtain oscillations in protein activity and abundance by

including production and degradation:

$$\frac{dX}{dt} = \varepsilon^{-1}(f(X)(X_T - X) - g(X)X)$$

$$\frac{dX_T}{dt} = k_X - X_T X. \tag{5}$$

As before, $k_X$ is the protein production rate and the dynamics of activation and inactivation are fast with respect to production and degradation, such that $\varepsilon$ is small. We assume that the active form of $X$ promotes its own degradation through mass-action kinetics (Fig 5A). Note that we have simplified the set of equations by omitting a term $(-X^2)$ from the first equation. In doing so, we ensure that the bistable response curve we have used before appears as a null-cline of the system. This simplification does not significantly change the system dynamics if $\varepsilon$ is small (see the Methods section). This system is analogous to the early embryonic cell cycle. There, Cdk1 activation brings about its own inactivation by degradation of the Cyclin B sub-units. This degradation in the cell cycle is mediated by APC/C, an interaction which is not present in our simple model.

For given functions $f$ and $g$, this system only oscillates for a specific range of $k_X$. If $k_X$ is too small, the production rate is not high enough to push the system over the activation threshold, and the system converges to a steady state with low activity. For a high value of $k_X$, degradation cannot compensate for production even when the protein is mostly active, and the system converges to a steady state with large $X$. For intermediate values of $k_X$, the system switches between accumulation and degradation with low and high $X$ respectively. This is illustrated in the phase plane in Fig 5B. If $k_X$ is such that the two nullclines intersect in between the two saddle-node points, the system converges to a stable limit cycle. This oscillation is marked by a slow increase along the bottom branch of the bistable curve, a slow decrease along the upper branch, and fast jumps in between (Fig 5B and 5C).

As before, we now allow the parameter $a$ to depend on the total amount of protein (Eq (4)). In the previous section we found that, in the presence of noise, transition times show less variation if the switch dynamically changes. In the case of oscillations with noise (Fig 5D), we find that more accurate transition times are reflected in a more stable period. A dynamic switch ($\Delta a > 0$) shows less variation in the period of the oscillation than a static switch (Fig 5E). Larger switch changes ensure smaller variation of the period (Fig 5F). The variation of periods is larger for extreme values of $k_X$, such that the nullclines intersect close to the saddle-node point. For those values, stochastic activation/inactivation is more likely, an effect which is mitigated by the dynamically changing switch. Similar results hold for a system in which the stochasticity is implemented using the Gillespie algorithm, however the effect is smaller (Fig 5G and S3 Fig). This is similar to what we found for the transition timing.

Not only the period, but also the amplitude shows less variation. However, the effect is less clear: dynamically changing the switch a little lowers the amplitude variation, but the effect does not persist for larger switch variations (S2 Fig).

In *Xenopus laevis*, early embryonic cycles have a remarkably stable period [76]. Combined with an initial period difference between different cells in the embryo, this shows as a wave of cell division. Dynamically changing the bistable switches in the early embryonic cell cycle would be one way to contribute to this observed stability in cell division period.

Next, we expand the model by including negative values of $\Delta a$, which corresponds to an activation threshold which increases as $X_T$ increases. Moreover, we model a biologically

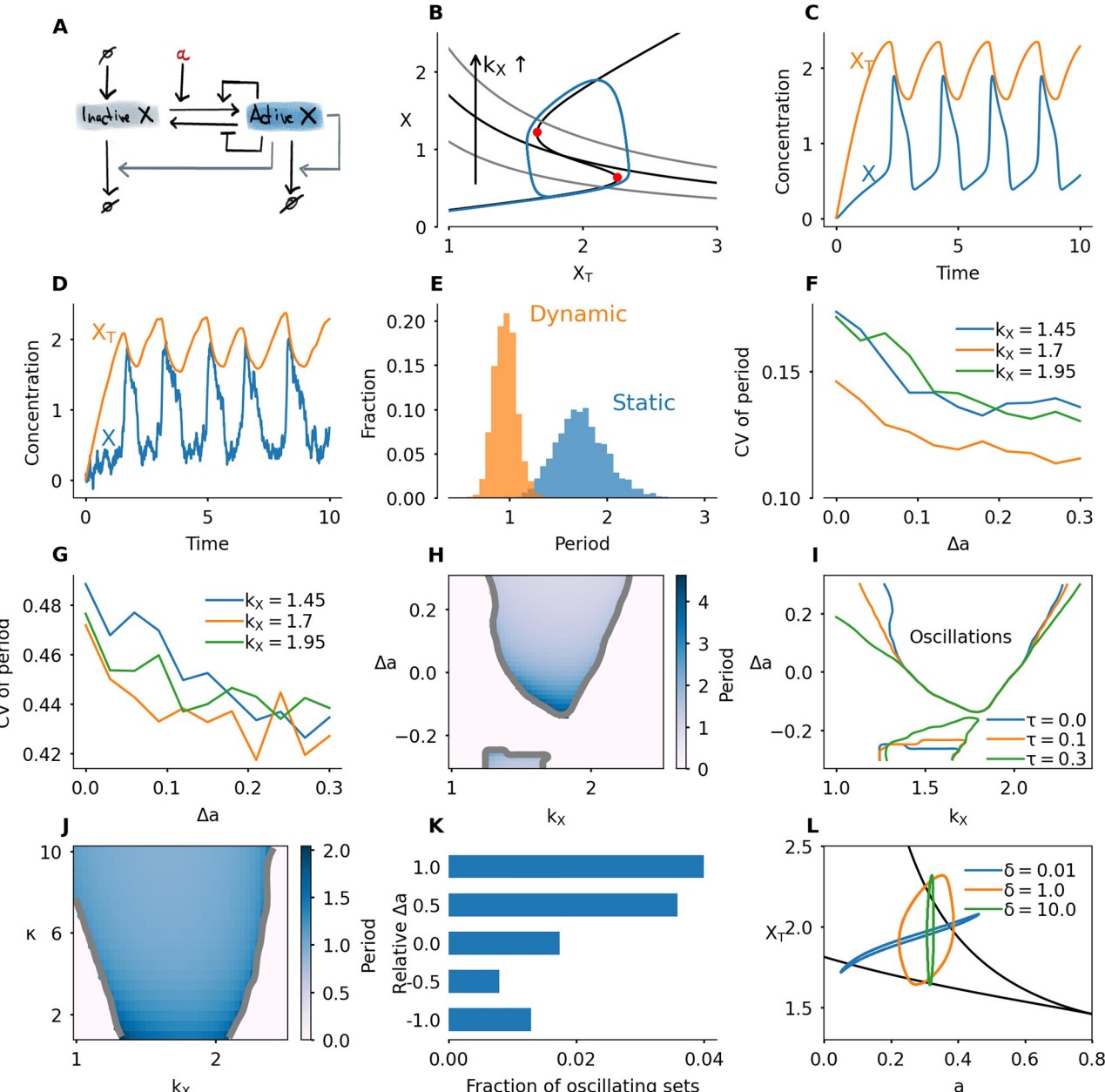

**Fig 5. Dynamic switches promote oscillations.** A) Interaction diagram. The active form of protein $X$ promotes its own degradation. B) Phaseplane of the system given by Eq 5 (static switch). The second nullcline depends on the value of $k_X$. The S-shaped nullcline has the same shape as the bistable response curve studied in the previous section. The second nullcline is given by $X = k_X/X_T$ and is shown for $k_X = 1.1$, 1.7 and 2.3. The blue limit cycle corresponds to $k_X = 1.7$. C) Time series of the oscillatory system with $k_X = 1.7$. D) Time series for the system with noise, $\sigma = 0.6$. E) Histogram of the period for a static ($\Delta a = 0$) and dynamic ($\Delta a = 0.3$) switch, with $\bar{a} = 0.3$, $\kappa = 5$, $\sigma = 0.6$, $k_X = 1.7$. F) Coefficient of variation (standard deviation divided by mean, CV) of the period in the oscillatory system with noise added to the $X$-variable. Here $\kappa = 5$, $\sigma = 0.6$. G) Coefficient of variation for the stochastic simulation algorithm, with $\Omega = 10$. H) Period in color, as function of $k_X$ and $\Delta a$ with $\kappa = 5$, $\tau = 0$, $\bar{a} = 0.3$. I) Oscillatory region in the ($k_X$, $\Delta a$)-plane for different values of the delay time $\tau$ with $\kappa = 5$, $\bar{a} = 0.3$. J) Period as function of $k_X$ and $\kappa$ with $\bar{a} = 0.3$, $\Delta a = 0.3$ and $\tau = 0.1$. K) Fraction of parameter sets for which the system oscillates for 10000 randomly sampled parameter sets. L) Three different limit cycles of the three equation system given by Eq 8. The parameter $\delta$ defines the timescale on which $a$ changes. See S6 Video for an animation corresponding to this panel. All parameters not described above can be found in the methods section, Table 1. For the Langevin version, statistics were obtained from runs for a total time of $T = 2000$ (Panels E,F). For the SSA version, the total run time was $T = 4000$ (Panel G).

plausible phase shift to the relation between $X_T$ and $a$ by including a time delay $\tau$:

$$a(t) = a(X_T(t - \tau)). \tag{6}$$

As mentioned before, oscillations occur if $k_X$ lies in a given interval. This interval becomes larger if $\Delta a$ increases, which indicates that larger changes of the bistable curve produce a larger region of oscillations (Fig 5H). This observation holds both with and without time delay, but the effect is larger if time delay is included (Fig 5I). The effect of increasing time delay is most pronounced for low $k_X$. Note that also for $\Delta a < 0$, there is an oscillatory region. When $\Delta a < 0$, the switch changes in the opposite direction of the change of $X_T$, i.e. the activation threshold moves to the right as $X_T$ approaches it. For small negative values of $\Delta a$, oscillations persist if the production rate is such that the system approaches the threshold fast enough while it moves to the right. The reappearance of oscillations for more negative values of $\Delta a$ is due to our implementation of the varying switch. These oscillations are qualitatively different: they are of lower amplitude and have less variation in the switch, see S5 Video.

The speed by which the bistable response curve changes also plays a role: faster, more abrupt transitions, which correspond to higher $\kappa$, promote oscillations (Fig 5J).

As a final demonstration of how dynamically changing switches affect the occurence of oscillations, we perform a random sampling of 10000 parameter sets. We sample all the parameters affecting the Hill functions $f$ and $g$, the timescale parameter $\varepsilon$, and $k_X$. For each parameter set we first detected whether the system is bistable, and if so, we simulated the model with $\Delta a = ia$, $i = -1, -1/2, 0, 1/2, 1$, for $\kappa = 1, 5, 10$ and $\tau = 0, 0.1, 0.2$. For each simulation we detect whether the system oscillates or goes into a steady state. Generally, oscillations are quite rare, but in all cases, having $\Delta a > 0$ increased the probability of obtaining oscillations (Fig 5K and S4 Fig). Note that oscillations can also exist for $\Delta a < 0$. In that case, however, oscillations are less likely.

To conclude, we have found that making the bistable response curve dynamic instead of static enhances the accuracy of the oscillation period in noisy systems, and increases the region in parameter space where oscillations are found. This effect is larger when the shape change lags the increase of $X_T$. S3 Video shows an oscillation with a dynamically changing bistable switch.

## 3 Discussion

In this paper, we have described how changing bistability in space and time can affect transitions, oscillations and propagation of fronts. We believe that our results are useful as a point of view which can be applied to different systems. In this discussion, we elaborate on potential applications and promising directions for future work.

Our main example has been mitotic entry, and the variables in the simplified model we used can roughly be mapped to the proteins and interactions involved at that transition. However, our results are conceptually valid at other biological transitions as well. The behavior we describe is generic in the sense that the only condition that is required is an increase of the "input" of the switch in combination with a variable affecting the activation threshold or the general shape of the switch. Such mechanisms are likely to be found at other cell cycle transitions as well.

The viewpoint of dynamic switches can be used to extend existing cell cycle models and interpret more complicated models. One direct way to increase the realism of cell cycle models is by including space. Even though numerous mathematical models of the cell cycle already exist, few of them include spatial regulation. Models that do include a spatial component often focus on traveling waves which can play a role in synchronizing large cells, such as *Xenopus* [71] or *Drosophila* embryos [74]. In *Xenopus* cell-free extracts, nuclei play an essential role as a pacemaker [72, 73], possibly due to the fact that nuclei locally increase concentrations of key

regulatory proteins [72]. This can possibly be interpreted as changing bistable response curves in nucleus and cytoplasm. In *Drosophila*, changes to the bistable switch have been proposed as an explanation of changing wavespeeds over different cycles [74], and bistable thresholds play a crucial role in the so-called sweep waves [75].

New insights are likely to be gained from models that also take into account the heterogeneity and spatial structures of a real cell. A first step towards that goal is to consider the compartments of nucleus and cytoplasm. Some cell cycle models have done this already (e.g. [77]) but we believe that exploring such models, possibly using the interpretation of changing bistability, provides a fruitful way to extend our knowledge into cell cycle regulation. This is especially the case since the importance of compartmentalization on dynamics has been seen both experimentally and in modeling studies in a number of different systems.

## Compartmentalization in the cell: Experimental and theoretical results

Compartmentalization can increase the complexity and richness of a biochemical system: by compartmentalizing chemical reactions, cells can locally increase concentrations to speed up reactions, or inversely keep certain reactions from happening by separating the reactants. The important additional regulation offered by compartmentalization has become a topic of interest to experimentalists and modelers alike. We give a few examples. In the cell cycle, Santos et al. [38] showed that spatial feedbacks are present in mitotic entry. This is one of the mechanisms we used in our biological example. Likewise in the cell cycle, Doncic et al. recently showed that compartmentalization of a bistable switch plays a role in the commitment point, also referred to as the *Start* checkpoint in yeast [78]. Nucleocytoplasmic shuttling has been found to play a key role in generating a switch-like response in the ERK signaling pathway [79, 80]. Another example is given by the circadian clock, whose period is determined by spatio-temporal regulation of CRY [81]. In addition, nucleocytoplasmic shuttling plays a role in the interaction between the proteins p53 and Per2. This interaction is involved in the coupling between the cell cycle and the circadian clock [82]. Finally, adding multiple compartments is also attempted in synthetic biological systems by introducing artificial membranes [83, 84].

From the mathematical standpoint, figuring out how spatial heterogeneity is best introduced poses an interesting challenge, where simplicity, computational efficiency and realism have to be weighed against one another. The method we used in the model of mitotic entry was ODE-based. For each compartment, every chemical species has its own ODE. This assumes that, inside a compartment, diffusion is fast and the system is well mixed.

Another option is to use fully spatial models consisting of partial differential equations (PDEs). Here, boundary conditions can be used to model fluxes across membranes between different compartments (see, e.g. [85, 86]), or the compartments can be introduced by setting the diffusion of some molecules to zero and initializing them in only one region of space (e.g. [87]). Yet another type of model is hybrid: some small compartments are considered to be well mixed, and are modeled by ODEs, whereas transport through the medium between the compartments is governed by a PDE (see e.g. [88] for a recent example of such a model in a biological system). If compartments are not bound by membranes, but instead generated by phase separation, modeling may need to take into account the physics of the phase separation process, a topic of current interest in cell biology [89].

## The role of different timescales

If fluxes between compartments are slow compared to typical activation/inactivation dynamics, the shape of the bistable response curve changes on a slow timescale. This corresponds to the implementation we used in this paper, where the change of the switch mediated by *a*

happens on similar timescales as the change in $X_T$. Our implementation—with the sigmoidal function $a(X_T)$ and the time lag $\tau$—is artificial but suitable for the message we want to convey. However, a more thorough theoretical study of such simple models could replace the explicit dependence of $a$ on $X_T$ by evolution equations for the dynamics of the switch shape, which can then be interpreted in the context of multiple-timescale systems. Such an extension of our model would take the form

$$\frac{dX}{dt} = \varepsilon^{-1}(f(X,a)(X_T - X) - g(X)X)$$

$$\frac{dX_T}{dt} = k_X - XX_T \tag{7}$$

$$\frac{da}{dt} = G(X, X_T, a),$$

where $G$ describes the dynamics of $a$. A concrete and simple example of such a model is given by

$$\frac{dX}{dt} = \varepsilon^{-1}(f(X,a)(X_T - X) - g(X)X)$$

$$\frac{dX_T}{dt} = k_X - XX_T \tag{8}$$

$$\frac{da}{dt} = \delta^{-1}(H(X_T) - a),$$

where $H(X_T) = \bar{a} + \Delta a \tanh(\kappa(X_T - X_c))$. In this system the variable $a$ relaxes to its value $H(X_T)$, and the scale on which this happens is governed by the parameter $\delta$. For $\delta \to 0$, this system reduces to our model with two equations, where $a$ follows $X_T$ according to the function $H$ (Eq 4). By modifying the parameter $\delta$, different kinds of dynamics can be obtained. In Fig 5L, we show some of the resulting limit cycles in the $(a, X_T)$ plane and S6 Video shows the effect of these timescales on the bistable switch.

Rigorous mathematical study of multiple-timescale systems has been done extensively in other areas, such as mathematical neuroscience [90, 91]. In stochastic models of chemical reactions, mathematical treatments of multiple-timescale systems are being developed too. There, the focus is often on ways to reduce the computational cost of a stochastic simulation using the Gillespie algorithm by separating reactions on different timescales. Developing such algorithms and assessing the conditions under which they provide good approximations has been the subject of many recent papers (see Refs. [92, 93] and references therein). In the case of our model, an expanded stochastic version might shed further light on the role of noise in the different variables.

Connecting a more rigorous mathematical analysis of these multiple-timescale systems to expected biological outcomes in the cell cycle can provide important clues to uncover the underlying dynamics. Moreover, this may lead to new opportunities for mathematicians and cell biologists to collaborate.

Finally, a note on how changing bistable switches might be observed experimentally. One of the outcomes of a mathematical model such as the one we studied here is a time series, which gives the evolution of concentrations of the main proteins over time. By performing a more detailed analysis, we can find out which qualitative features are specific to a time series derived from changing bistability. Next, experiments can be set up to try to detect such features. Another approach would be to measure the steady-state response curves to obtain activation thresholds, and perform this experiment under different experimental conditions.

## 4 Conclusion

Bistable switches play a crucial role in the cell cycle, providing a mechanism for quick and irreversible transitions. They also lie at the basis of more complex behavior such as spatial front propagation and relaxation oscillations. The classic viewpoint of a static switch and fixed activation and inactivation thresholds does not take into account that the factors that determine the shape of the response curve can vary over time. In a biochemical system, these factors are typically protein concentrations. When those concentrations change, the bistable response curve changes, and this happens all while the system is proceeding along the branches of the bistable curve.

We showed that such changing bistability can arise naturally in a biological system, using nucleus and cytoplasm as compartments in a model of mitotic entry. There, spatial translocation of proteins allows additional control over activation thresholds, which may differ in different compartments. Next, we examined the consequences of a dynamically changing switch in a simple model. We have shown that such a mechanism allows more accurate control of the transition timing in noisy systems, if the noise in the slow variable is negligible compared to the noise in the fast variable. Additionally, by controlling protein levels in space, the location and speed of propagating fronts of activity can be regulated. In oscillatory systems, a changing bistable switch increases the robustness of the oscillations to parameter variations. The control of transition timing and avoiding a premature transition can play a role in cell cycle checkpoints, whereas more robust oscillations may be important in early embryonic cell cycles which behave like autonomous oscillators. These advantages suggest that such dynamic regulation may have evolved as an extra mechanism in the cell's repertoire to ensure faithful genome replication and division. Accuracy and robustness are probably not the only advantages of a changing switch. For example, if there is an energetic cost associated to maintaining a bistable switch at a certain level, dynamically controlling the shape can be a way to use energy more efficiently. This kind of temporal compartmentalization is widely seen in biology. Circadian rhythms, for example, provide a means of compartmentalizing processes to align with external light and temperature cycles and therefore optimally use energy [94]. This energy-based view of dynamic bistable switches will perhaps benefit from a thermodynamic description, which takes into account energy consumption (e.g. [95]).

Mathematical modeling and concepts derived from nonlinear dynamics, such as bistability and limit cycles, have been very influential on our understanding of many biological phenomena, and will continue to be, as has been recently advocated by Tyson and Novák [96]. In our discussion we have echoed some of their perspectives. Additionally, in this paper, by adding an extra layer to the regulation of cell cycle transitions we have attempted to push our dynamical understanding of this fundamental process a little bit further.

Even though the simple model we studied in this paper was artificial, its main conclusions will likely hold for more realistic mathematical models. In fact, we suspect that a dynamically changing bistable switch is already present in many published models, but not recognized or described as such. Therefore, we propose that the dynamically changing switches we described can be used as a means to interpret existing models and perhaps inspire new ones.

## 5 Methods

### 5.1 Standard parameter set

In all of our simulations, except for the ones done to obtain Fig 5K, we have used the same values for the parameters that describe the positive and negative feedback functions $f$ and $g$ (Eq 2), except for the parameter $a$ which, in the case of a dynamic switch, is changing in time. Table 1 gives a list of these parameters. The parameters used for obtaining the changing switch

**Table 1. Standard parameter values used for the feedback functions.**

| Parameter | Value |
|-----------|-------|
| $b$ | 1 |
| $K$ | 1 |
| $n$ | 5 |
| $a'$ | 0.1 |
| $b'$ | 1 |
| $K'$ | 1 |
| $m$ | 5 |
| $\varepsilon$ | 0.05 |

$\Delta a$, $\kappa$ were varied, the values are mentioned in the captions of the relevant figures. The production parameter $k_X$ varies over simulations and is also mentioned in the captions. The timescale parameter $\varepsilon$ was taken to be 0.05. For simulations with a delay time $\tau$, its value is mentioned in the caption.

The threshold $X_c$, which is used in the function that defines how the parameter $a$ varies with the total concentration $X_T$, is always taken to be the middle of the horizontal coordinates of the saddle-node points of the static bistable switch.

## 5.2 Sofware and algorithms

Simulations, data analysis, plotting and animations were all done in Python except for the two-parameter bifurcation diagram in Figs 3D and 5L, which was created using the interface to AUTO of the software XPPAUT [97]. The majority of our simulations use ODEs, but some versions of the model are delay differential equations or stochastic differential equations. To simulate all of these, we made use of the software packages JITCODE, JITCDDE and JITCSDE for Python, which implement solvers for ordinary, delay and stochastic differential equations respectively [98]. For parameter sweeps we used a high-perfomance computing cluster.

To obtain the bistable response curve, we implemented a pseudo-arclength continuation algorithm directly in Python (see, e.g., [99]). This gave us the flexibility to compute response curves on the fly, as for the animations.

For the period detection in the oscillatory systems, we start from the time series $X(t)$ and detect the times $t_{u,i}$ and $t_{d,i}$ when $X$ crosses a certain threshold up and a certain threshold down. The threshold up is the vertical coordinate of the left saddle-node point of the static switch, the down threshold is the vertical coordinate of the rightmost saddle-node point. Next, we remove repeated up or down crossings from this list, such that we obtain a list of alternating up and down crossing times. Finally we compute the period as the differences $t_{u,i+1} - t_{u,i}$. In the noisy system, this gives a set of period $P_i$ on which we can perform statistics. For the deterministic systems this value is constant. This method ensures that we only track oscillations that go around both branches of the bistable system, i.e. of sufficient amplitude.

To determine the amplitude (S2 Fig), we first smooth the time series using a low-pass Butterworth filter (using scipy's function `butter` and `sosfiltfilt`) and then detect minima and maxima. The amplitude is then the difference between a maximum and the following minimum.

To detect the contours of the oscillatory regions in Fig 5H–5J, we detected all points in the heatmap where the period goes from zero to positive and then applied smoothing. Note that this boundary is not strictly the same as the boundary between steady state and oscillations, since we consider only oscillations of sufficient amplitude, that go around both branches of the switch.

## 5.3 Stochastic models

We use two types of stochastic model: one stochastic differential equation model, of Langevin type, and one fully stochastic model in which we use Gillespie's algorithm [65] for simulation.

The Langevin equation we use is

$$
\begin{aligned}
dX &= \varepsilon^{-1}(f(X,a)(X_T - X) - g(X)X)dt + \sigma dW \\
dX_T &= k_X dt.
\end{aligned}
\tag{9}
$$

We include noise only in the fast variable, which is not a correct representation of molecular noise, but the simplest way to extend our ODE model to include stochasticity. We include noise only in the fast variable to simplify the system and only allow transition through "vertical" deviations from the steady state branch, in line with typical studies on stochastic switching [100]. Moreover, the ratio of noise magnitude in fast and slow variable is high when stochastic differential equations are derived from a discrete stochastic model [101], [93, Sec. 4.1], such that setting the noise on the slow variable to zero is reasonable. To determine transition times, we detect the timepoint when $X$ crosses a threshold concentration. This threshold concentration is always the average vertical coordinate of the saddle-node points of the static bistable switch.

For the fully stochastic model with discrete molecule numbers, let the number of molecules of the active protein $X$ be called $n_X$, and $n_Y$ denotes the number of inactive molecules $Y$. This means that the total amount of protein is $n_X + n_Y$. In order to compare the results of the stochastic simulation to the results of the deterministic model, we introduce a system size $\Omega$ and interpret the variables of Eqs 1, 3 and 5 as concentrations. To transfer the rates and parameters used in the deterministic system to the stochastic system, we multiply or divide the appropriate parameters by $\Omega$: any parameter which would have units of concentration in the deterministic model is multiplied by $\Omega$, parameters with unit of 1/concentration are divided by $\Omega$. See e.g. Ref. [93] for a thorough explanation of the simulation method and the translation from deterministic rate equations to the corresponding stochastic system.

The possible reactions in the model are given in Table 2. In the simulations used for Fig 4, i.e. without degradation, reactions 4 and 5 are omitted.

Here $f$ and $g$ are the Hill functions used before, now modified such that the threshold corresponds to a molecule number:

$$
\begin{aligned}
f(n_X) &= a + b\frac{n_X^n}{(\Omega K)^n + n_X^n} \\
g(n_X) &= a' + b'\frac{(\Omega K')^m}{(\Omega K')^m + n_Y^m}.
\end{aligned}
$$

**Table 2. Reactions in the stochastic model.**

|  | Reaction | Description | Rate |
|---|---|---|---|
| R1 | $Y \rightarrow X$ | Activation | $\frac{1}{\varepsilon}f(n_X,a)n_Y$ |
| R2 | $X \rightarrow Y$ | Inactivation | $\frac{1}{\varepsilon}g(n_X)n_X$ |
| R3 | $\varnothing \rightarrow Y$ | Production | $k_X\Omega$ |
| R4 | $X + Y \rightarrow \varnothing$ | Degradation of inactive protein | $n_X n_Y/\Omega$ |
| R5 | $X + X \rightarrow \varnothing$ | Degradation of active protein | $n_X(n_X - 1)/\Omega$ |

As before, for changing switches, $a$ depends on the total amount of protein:

$$a = \bar{a} + \Delta a \tanh\left(\frac{\kappa}{\Omega}\left(n_X + n_Y - (\Omega X_c)\right)\right).$$

All the parameters of the Hill functions as well as the threshold for the changing switch are the same as those used in the deterministic system (Table 1). We use Gillespie's direct method to simulate the system [65].

## 5.4 Removal of degradation term in the *X*-equation

In the oscillatory system, $X$ degrades itself, both in active and inactive form. The set of equations corresponding to this is

$$\frac{dX}{dt} = \varepsilon^{-1}(f(X)(X_T - X) - g(X)X) - X^2$$

$$\frac{dX_T}{dt} = k_X - X_T X. \tag{10}$$

We can rewrite this as

$$\frac{dX}{dt} = \varepsilon^{-1}(f(X)(X_T - X) - (g(X) + \varepsilon X)X)$$

$$\frac{dX_T}{dt} = k_X - X_T X, \tag{11}$$

and since $\varepsilon$ is considered to be small, the shape of the bistable switch induced by these equations is nearly the same as that induced by the one where $\varepsilon = 0$, which we use.

## 5.5 Spatial model

For the simulations in space, we use the equations

$$\frac{\partial X}{\partial t} = D_X \frac{\partial^2 X}{\partial x^2} + \varepsilon^{-1}(f(X)Y - g(X)X)$$

$$\frac{\partial Y}{\partial t} = D_Y \frac{\partial^2 Y}{\partial x^2} - \varepsilon^{-1}(f(X)Y - g(X)X). \tag{12}$$

here, $Y$ is the inactive form of the protein. These equations allow more flexibility in choosing, for example, different diffusion constants for active and inactive form. We took $D_X = D_Y = 5$ in our simulations for Fig 4K and 4L. We simulated these equations using a forward difference in time and centered difference for the space derivative. We use zero-flux boundary conditions. For Fig 4K, we detect the front position as function of time and fit a linear function. Initial conditions are always a step function.

## 5.6 Sampling of parameters

Sampling of the parameter sets in Fig 5K was done as follows: each parameter was sampled uniformly and independently from a given interval. For $\varepsilon$ we sampled the logarithm. S2 Table shows the intervals and whether the parameter was sampled logarithmically or not. The system was simulated with sampled parameters for a total time of $T = 200$. We considered a set of parameters as oscillatory if the period is larger than 0.01.

### 5.7 Full set of equations for the biological example

Here we give the details of the equations used to simulate mitotic entry as described in Section 2.1. We keep track of three different variables: total Cyclin B-Cdk1 complexes ([Cyc]), active Cyclin B-Cdk1 complexes ([Cdk1]) and Cdc25 levels ([Cdc25]). The model equations are based on the equations used by Yang and Ferrell [60]. Note that Cdc25 is a scaled variable: the value 1 would correspond to the level assumed by Yang and Ferrell. Each variable has a nuclear and cytoplasmic version which is denoted by subscript $n$ or $c$. Cyclin B is constantly produced at a rate $k_s$ and binds immediately to Cdk1 to create active Cdk1-Cyclin B complexes, hence the same production term $k_s$ in the equations for [Cyc] and [Cdk1]. We assume that production only happens in the cytoplasm. The activation rate of Cdk1 depends on Cdc25 levels and activity. The level is controlled by the variable [Cdc25], the activity is a function of Cdk1, since Cdk1 is an activator of Cdc25. The inactivation rate of Cdk1 depends on Wee1 levels and activity. We assume that total Wee1 levels are constant, but this level is higher in the nucleus than in the cytoplasm. In the simulation used in Fig 2, we used $[\text{Wee1}]_n = 1.3$ and $[\text{Wee1}]_c = 1$. This variable is also scaled, $[\text{Wee1}] = 1$ corresponding to the model used by Yang and Ferrell. As initial conditions for Cdc25, we use $[\text{Cdc25}]_n = 1$, $[\text{Cdc25}]_c = 2$.

Cyclin B-Cdk1 complexes and Cdc25 can be imported and exported from the nucleus with certain import and export rates. We use the convention that the subscript $n$ or $c$ for the rate denotes the compartment towards which the protein is moved. To account for the observation that both Cyclin B-Cdk1 and Cdc25 import is increased at mitotic entry, we introduce the functions $I_{\text{Cyc}}$ and $I_{\text{Cdc}}$, which modify the import rates of Cyclin B-Cdk1 and Cdc25 respectively. We use

$$I_{\text{Cyc}}([\text{Cdk1}]_n) = 0.1 + \frac{1}{30}[\text{Cdk1}]_n \tag{13}$$

$$I_{\text{Cdc}}([\text{Cdk1}]_c) = 1 + \frac{1}{60}[\text{Cdk1}]_c. \tag{14}$$

The equations are

$$\frac{d[\text{Cyc}]_n}{dt} = k_{n,\text{Cyc}}I_{\text{Cyc}}([\text{Cdk1}]_n)[\text{Cyc}]_c - k_{c,\text{Cyc}}[\text{Cyc}]_n \tag{15}$$

$$\frac{d[\text{Cyc}]_c}{dt} = -k_{n,\text{Cyc}}I_{\text{Cyc}}([\text{Cdk1}]_n)[\text{Cyc}]_c + k_{c,\text{Cyc}}[\text{Cyc}]_n + k_s \tag{16}$$

$$\frac{d[\text{Cdk1}]_n}{dt} = k_{n,\text{Cyc}}I_{\text{Cyc}}([\text{Cdk1}]_n)[\text{Cdk1}]_c - k_{c,\text{Cyc}}[\text{Cdk1}]_n \tag{17}$$

$$+[\text{Cdc25}]_n\left(a_{\text{Cdc25}} + b_{\text{Cdc25}}\frac{[\text{Cdk1}]_n^{m_{\text{Cdc25}}}}{K_{\text{Cdc25}}^{m_{\text{Cdc25}}} + [\text{Cdk1}]_n^{m_{\text{Cdc25}}}}\right)([\text{Cyc}]_n - [\text{Cdk1}]_n) \tag{18}$$

$$-[\text{Wee1}]_n\left(a_{\text{Wee1}} + b_{\text{Wee1}}\frac{K_{\text{Wee1}}^{m_{\text{Wee1}}}}{K_{\text{Wee1}}^{m_{\text{Wee1}}} + [\text{Cdk1}]_n^{m_{\text{Wee1}}}}\right)[\text{Cdk1}]_n \tag{19}$$

$$\frac{d[\text{Cdk1}]_c}{dt} = -k_{n,\text{Cyc}}I_{\text{Cyc}}([\text{Cdk1}]_n)[\text{Cdk1}]_c + k_{c,\text{Cyc}}[\text{Cdk1}]_n + k_s \tag{20}$$

$$+[\text{Cdc25}]_c \left( a_{\text{Cdc25}} + b_{\text{Cdc25}} \frac{[\text{Cdk1}]_c^{m_{\text{Cdc25}}}}{K_{\text{Cdc25}}^{m_{\text{Cdc25}}} + [\text{Cdk1}]_c^{m_{\text{Cdc25}}}} \right) ([\text{Cyc}]_c - [\text{Cdk1}]_c) \tag{21}$$

$$-[\text{Wee1}]_c \left( a_{\text{Wee1}} + b_{\text{Wee1}} \frac{K_{\text{Wee1}}^{m_{\text{Wee1}}}}{K_{\text{Wee1}}^{m_{\text{Wee1}}} + [\text{Cdk1}]_c^{m_{\text{Wee1}}}} \right) [\text{Cdk1}]_c \tag{22}$$

$$\frac{d[\text{Cdc25}]_n}{dt} = k_{n,\text{Cdc25}} I_{\text{Cdc}}([\text{Cdk1}]_c)[\text{Cdc25}]_c - k_{c,\text{Cdc25}}[\text{Cdc25}]_n \tag{23}$$

$$\frac{d[\text{Cdc25}]_c}{dt} = -k_{n,\text{Cdc25}} I_{\text{Cdc}}([\text{Cdk1}]_c)[\text{Cdc25}]_c + k_{c,\text{Cdc25}}[\text{Cdc25}]_n \tag{24}$$

The parameters can be found in S1 Table and are mostly taken from [60]. The import rates and the functions that influence those rates were chosen to obtain a good example of the mechanism we propose.

## Supporting information

**S1 Fig. Transition timing: Standard deviation, mean and coefficient of variation for the stochastic simulation algorithm.**
(PDF)

**S2 Fig. Standard deviation, mean and coefficient of variation of the amplitude of oscillations, Langevin equation. Here $\sigma = 0.6$, $\kappa = 5$.**
(PDF)

**S3 Fig. Standard deviation, mean and coefficient of variation of the period for different system sizes $\Omega$.**
(PDF)

**S4 Fig. Results of the random parameter sampling for different values of $\tau$ and $\kappa$.**
(PDF)

**S1 Table. Parameter values used for the model of mitotic entry.** All parameters except for the import and export rates and $k_s$ were taken from [60].
(PDF)

**S2 Table. Range from which parameters are sampled.** For $\varepsilon$, we sampled the logarithm: we sampled a number $\gamma$ between $\ln(0,01)$ and $\ln(1)$ uniformly and then took $\varepsilon = e^\gamma$.
(PDF)

**S1 Video. Transition with a changing switch.** This animation corresponds to Fig 4C. The protein is produced at a constant rate, while the activation threshold is moving to the left due to an increase in $a$. The effect is a fast transition, and $X$ activity stays low until the transition.
(MP4)

**S2 Video. Moving front with redistribution of $a$.** This animation illustrates the blocking of the front due to a heterogeneity in $a$, and the release of the front due to redistribution of $a$ (Fig 4K and 4L). The $a$ profile we use is a smooth hyperbolic tangent function of $x$. The high value of $a$ is 0.27, the low value is 0.13. The front gets stuck at the transition to low $a$, where $a \approx 0.15$. At time $t = 25$, we effect a smooth transition to the flipped $a$-profile which releases the front, after which it continues moving to the right.
(MP4)

**S3 Video. Oscillation with a changing switch.** This animation illustrates a system with production and degradation. We took $k_X = 1.7$, $\bar{a} = 0.3$, $\Delta a = 0.2$, $\kappa = 5$, $\tau = 0$. The bistable switch is changing while the system oscillates.
(MP4)

**S4 Video. Mitotic entry with two compartments.** This animation corresponds to Fig 2 in the main text. At first, Cyclin B accumulates in the cytoplasm. The activation threshold for Cdk1 is lower there, so Cdk1 activity jumps to the upper branch first in the cytoplasm. This triggers nuclear import of Cdc25, which lowers the threshold in the nucleus. Following this, Cdk1 activity in the nucleus jumps up, which triggers an increased import of Cyclin B. Cdk1 activity in the nucleus keeps increasing while in the cytoplasm it settles to a constant value.
(MP4)

**S5 Video. Oscillations for negative Δ$a$.** This animation shows what the oscillations for negative Δ$a$ look like which appear in Fig 5H. The animation shows oscillations in the phaseplane for different Δ$a$ with $k_X = 1.5$ and $\tau = 0$. On the left, small amplitude oscillations exist for very negative Δ$a$, as a consequence of our implementation of the varying switch. Second plot: no oscillations exist here, the activation threshold moves to the right and the system is stuck on the lower branch. Third plot: small negative Δ$a$: large amplitude oscillations exist: even though the activation threshold moves to the right, the system manages to cross it. Rightmost plot: positive Δ$a$, large amplitude oscillations in which the activation threshold moves to the left as the system approaches it. In the phase planes, the grey response curve corresponds to the value of $\bar{a}$, around which the switch varies. As can be seen, in the leftmost plot the switch does not actually vary around this curve, but stays on the left of it.
(MP4)

**S6 Video. Oscillations in a three equation model with different timescales.** This animation corresponds to Fig 5L. We show the evolution of the system given by Eq 8, with three different values of $\delta$. For $\delta = 0.01$, $a$ relaxes very quickly to the value given by $H(X_T)$, which produces the curve traced out in the $(a, X_T)$ plane. This situation is very close to the two-equation system we use in the rest of the paper. For $\delta = 1$, $a$ relaxes more slowly to $H(X_T)$. The limit cycle traced out in the $(a, X_T)$ plane is more open, and the oscillation does not follow the bistable switch very closely. Finally for $\delta = 10$, the dynamics of $a$ are so slow that it stays nearly constant, because the oscillations of $X_T$ are "averaged out". This corresponds to a nearly vertical projection of the limit cycle in the $(a, X_T)$-plane.
(MP4)

## Acknowledgments

We are grateful to the members of the Gelens lab for useful comments on the manuscript.

## Author Contributions

**Conceptualization:** Jan Rombouts, Lendert Gelens.

**Data curation:** Jan Rombouts.

**Formal analysis:** Jan Rombouts.

**Funding acquisition:** Jan Rombouts, Lendert Gelens.

**Investigation:** Jan Rombouts.

**Methodology:** Jan Rombouts.

**Project administration:** Lendert Gelens.

**Resources:** Lendert Gelens.

**Software:** Jan Rombouts.

**Supervision:** Lendert Gelens.

**Validation:** Jan Rombouts.

**Visualization:** Jan Rombouts.

**Writing – original draft:** Jan Rombouts, Lendert Gelens.

**Writing – review & editing:** Jan Rombouts, Lendert Gelens.

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
