## [Decision Letter · Decision Letter 0]

7 Sep 2020

Dear Dr. Gelens,

Thank you very much for submitting your manuscript "Dynamic bistable switches enhance robustness and accuracy of cell cycle transitions" for consideration at PLOS Computational Biology.

As with all papers reviewed by the journal, your manuscript was reviewed by members of the editorial board and by several independent reviewers. In light of the reviews (below this email), we would like to invite the resubmission of a significantly-revised version that takes into account the reviewers' comments.

We cannot make any decision about publication until we have seen the revised manuscript and your response to the reviewers' comments. Your revised manuscript is also likely to be sent to reviewers for further evaluation.

Sincerely,

Attila Csikász-Nagy

Associate Editor

PLOS Computational Biology

Mark Alber

Deputy Editor

PLOS Computational Biology

Reviewer's Responses to Questions

**Comments to the Authors:**

Reviewer #1: This is nice work that is entirely suitable for publication in PLoS Comp Biol.

The authors have already made most of the changes that I recommended in an earlier review of the paper for a different journal.

A necessary revision: the authors should create a table that lists all the parameter values used in the calculations in the figures. For example, I wanted to check Fig. 3, and it took me a little bit of trial-and-error to figure out the parameter values used for the model in this figure.

An optional revision: In the Discussion, the authors say "It would be valuable to study a model of the form... Eq. (7)". They don't have to put this off to a future publication, because they are already set up to do so using the two-param bifurcation diagram in Fig 3D. In the attached file, I present an implementation of Eq. (7) that produces limit cycle oscillations. When XT(t) and a(t) are projected onto Fig. 3D, the limit cycle sweeps diagonally across the bistable domain (the blue region), alternately flipping X(t) into the high and low states of the bistable switch. It would be a nice example of how to analyze a system with time-scale separations.

Reviewer #2: The idea presented in this manuscript is very interesting, but I think it can be improved:

- The introduction is broad and it doesn't focus on the main ideas stated in the abstract. It would be nicer to have more details about dynamical stability than explain bistable switches in the cell cycle... which they have been broadly covered.

- At the end of section 2.1, there is explained that the NEBD is another example, but it is not supported by any analysis or bibliography covering the dynamical changes on bistable switches when including NEBD.

- Fig 2C. It is difficult to follow together with the description in the main test. It would be nice to have some kind of guidance when walking through the different plots within the test. Besides, the colours are misleading, as one can think that blue curve is the complex cycB-CDK1 and Cdc25 is red (based on the diagram on Fig 2A). Choose another colour scheme would be helpful .

- Section 2.2. seems completely irrelevant for the purpose of the manuscript. The system explained here can de added into the next 2 sections, as it seems to be the original work described in the manuscript.

- Apart from section 2.1, there is no extra connection with cell cycle, either in the switches or the oscillatory behaviour. The toy examples used to explain the idea are quite useful, but I am missing a relation between these examples and the real players of the cell cycle (out of the brief explanation on the discussion).

- The title of the work oversells the content of the manuscript. If kept, I will consider to make a strong point based on the previous comment.

Reviewer #3: attached

**Have all data underlying the figures and results presented in the manuscript been provided?**

Reviewer #1: **No: **Must give param values so readers can reproduce all figures in the paper.

Reviewer #2: Yes

Reviewer #3: Yes

PLOS authors have the option to publish the peer review history of their article (what does this mean?). If published, this will include your full peer review and any attached files.

Reviewer #1: No

Reviewer #2: No

Reviewer #3: No
---

## [Decision Letter · Decision Letter 1]

14 Nov 2020

Dear Dr. Gelens,

Thank you very much for submitting your manuscript "Dynamic bistable switches enhance robustness and accuracy of cell cycle transitions" for consideration at PLOS Computational Biology. As with all papers reviewed by the journal, your manuscript was reviewed by members of the editorial board and by several independent reviewers. The reviewers appreciated the attention to an important topic. Based on the reviews, we are likely to accept this manuscript for publication, providing that you modify the manuscript according to the review recommendations.

Sincerely,

Attila Csikász-Nagy

Associate Editor

PLOS Computational Biology

Mark Alber

Deputy Editor

PLOS Computational Biology

[LINK]

Reviewer's Responses to Questions

**Comments to the Authors:**

Reviewer #2: The authors have addressed most of my suggestions properly. The ones that were not addressed were reasoned accordingly.

Reviewer #3: The authors have addressed most of concerns in the revision. However, according to the new result, the description for the main results needs to be more clear. The authors newly performed the stochastic simulations for the case when the slow variable also fluctuates (Fig 4I, J, Fig 5G, and Fig S1 and S3). According to these new results, the advantage of dynamic switch overs the static switch is not clear. Even dynamic switch leads to more unstable oscillations than the static switch when Omega=50 and 100 (e.g. red and green line; Fig S3 left lowest panel). Thus, the current results indicate that the dynamic switch leads to robustness (stability) over the static switch “under the condition that the noise of the slow variable, which leads to the dynamic switch (i.e. XT), is negligible”. However, such limitation seems not clearly descried in the current text including intro and conclusion. Providing the clear condition when the dynamic switch is robust will improve the impact of the manuscript and be more helpful for readers when they want to use the results.

**Have all data underlying the figures and results presented in the manuscript been provided?**

Reviewer #2: Yes

Reviewer #3: Yes

PLOS authors have the option to publish the peer review history of their article (what does this mean?). If published, this will include your full peer review and any attached files.

Reviewer #2: No

Reviewer #3: No
---

## [Editor Report · Decision Letter 2]

18 Nov 2020

Dear Dr. Gelens,

We are pleased to inform you that your manuscript 'Dynamic bistable switches enhance robustness and accuracy of cell cycle transitions' has been provisionally accepted for publication in PLOS Computational Biology.

Best regards,

Attila Csikász-Nagy

Associate Editor

PLOS Computational Biology

Mark Alber

Deputy Editor

PLOS Computational Biology

---

## [Editor Report · Acceptance letter]

22 Dec 2020

PCOMPBIOL-D-20-01411R2 

Dynamic bistable switches enhance robustness and accuracy of cell cycle transitions

Dear Dr Gelens,

I am pleased to inform you that your manuscript has been formally accepted for publication in PLOS Computational Biology. Your manuscript is now with our production department and you will be notified of the publication date in due course.

With kind regards,

Jutka Oroszlan
